# Memory-Enhanced Temporal Learning: Leveraging SAM2's Memory Modules for Consistent Segmentation on Surgical Video

Shunsuke Kikuchi[1,2], Atsushi Kouno[1], and Hiroki Matsuzaki[1]

[1] Jmees Inc., Kashiwa-city, Chiba, Japan https://www.jmees-inc.com/en/
[2] Computational and Systems Biology Program, UCLA, Los Angeles CA 90095, USA
`shunkikuchi2111@ucla.edu`

**Abstract.** Video segmentation is critical for many medical imaging applications; however, developing video-aware models is challenging as they require densely annotated large-scale datasets. Most mainstream segmentation models process each frame independently, often resulting in inconsistent segmentation masks across consecutive frames. Although the recently proposed Segment Anything Model 2 (SAM2) has demonstrated promising segmentation capabilities with its memory mechanism, applying SAM2 in clinical settings is challenging due to its reliance on user prompts.

To address these issues, we introduce the Temporal Memory Augmentation Module (TMAM). TMAM adapts any pre-trained 2D segmentation model by encoding past-frame predictions via SAM2's memory encoder and applying memory attention to refine current-frame features. By leveraging temporal redundancy in video sequences, TMAM captures contextual cues that may be overlooked by single-frame processing, thereby improving robustness to occlusions and boundary artifacts.

Experiments on public surgical video datasets demonstrate that TMAM enhances Dice scores and temporal consistency across various base architectures. These results highlight TMAM's ability to produce smoother, more coherent segmentations, paving the way for more reliable video analysis in surgical image navigation systems and robotic surgery, where precise and consistent segmentation is essential.

**Keywords:** Video Segmentation · Endoscopic Video · Temporal Consistency.

## 1 Introduction

Accurate and temporally consistent video segmentation is crucial in medical image analysis, particularly for high-stakes applications such as surgical support systems and robotic surgery. In these settings, reliable segmentation not only enhances the precision of tissue and instrument tracking but also directly contributes to improved patient outcomes. Recent advances, exemplified by the Segment Anything Model 2 (SAM2), have demonstrated that incorporating memory

mechanisms to capture temporal context can dramatically enhance segmentation performance [11].

Based on recent trends in MICCAI EndoVis Challenges [10, 13, 18], the main approach for organ and instrument segmentation has been to apply frame-by-frame segmentation models to videos. This method tends to yield stable performance, particularly in challenging and data-scarce medical imaging scenarios. However, this approach fails to leverage the rich temporal continuity inherent in video data—an issue that is particularly pronounced in medical videos, where occlusions, rapid motion, and partial object appearances are common, and temporally dense annotations are rarely available.

On the other hand, video-specific segmentation methods have been proposed [2, 5, 8]. Yet, adapting such models to the medical domain is nontrivial; high training costs, the need for temporally dense annotations, and prolonged training times pose significant challenges. Moreover, while considerable progress has been made in video segmentation for natural images, methods tailored for medical videos—characterized by subtle anatomical differences and limited data—remain sparse.

Motivated by these challenges, we propose the **Temporal Memory Augmentation Module (TMAM)**. By leveraging transfer learning, TMAM reduces the cost of learning temporal information and effectively extracts and integrates temporal redundancy from even sparsely annotated medical video datasets. TMAM transfers SAM2's memory encoder and memory attention mechanisms to enhance existing 2D segmentation architectures. The key contributions of our paper are:

1. By integrating past-frame predictions through a memory encoder and refining current-frame features via memory attention, our approach mitigates inconsistencies caused by occlusions and boundary artifacts while significantly improving temporal coherence. It is expected to provide stable boundaries in long surgical videos, resulting in easily interpretable masks for humans and noise free for the surgical system.

2. The TMAM is designed as a plug-and-play module compatible with all encoder-decoder segmentation architectures, seamlessly enabling temporal inference capabilities without extensive modifications.

3. The TMAM leverage a domain-specific encoder during the frame propagation process, which is not possible with the standard SAM2 framework. This allows TMAM to recover from initial prediction errors and robustly handle challenges like occlusions

4. Furthermore, regarding automation, a simple tracking approach based on SAM2 would fail as the video progresses. It cannot predict instruments or organs that appear after the initial frame. Requiring new prompts for every new object during a surgical procedure is impractical. TMAM solves this by enabling true end-to-end automatic segmentation.

## 2    Methods

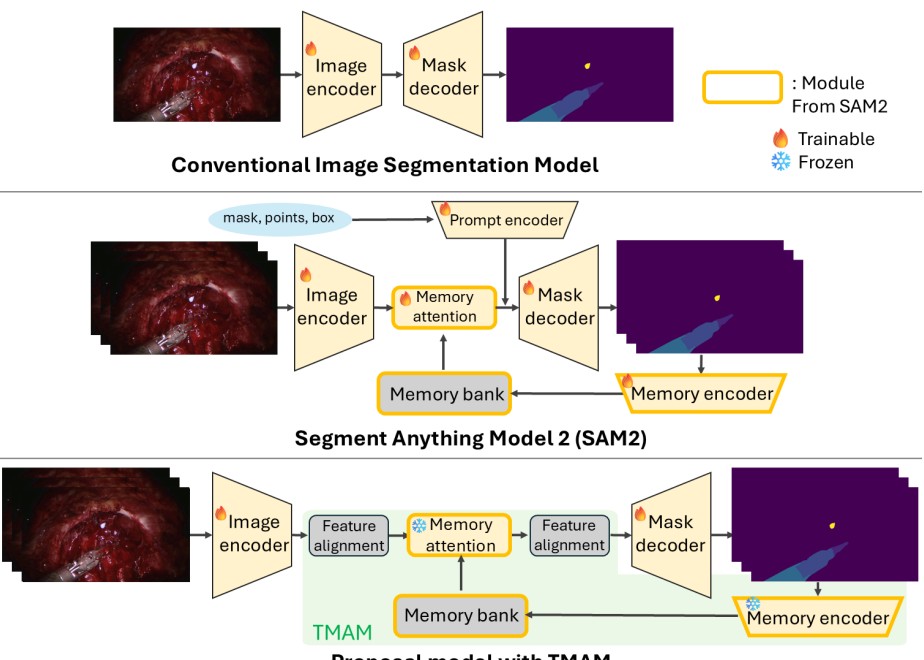

**Fig. 1.** Comparison of Conventional segmentation model, SAM2, and our proposed model incorporates a Temporal Memory Alignment Module (TMAM) to enhance feature consistency and segmentation accuracy. Trainable and frozen components are indicated.

**Overall Framework.** Figure 1 illustrates the architecture of TMAM and its functionality. TMAM is designed to integrate with any pretrained 2D segmentation model consisting of an encoder and decoder. It incorporates the memory encoder and memory attention components from SAM2.1-base, performing feature alignment to adjust tensor sizes and dimensions. With this alignment, TMAM performs segmentation without the prompts required in SAM2. Additionally, TMAM employs an internal memory module that stores temporal representations from previous frames, which are retrieved during memory attention to refine current-frame predictions.

**Memory Attention.** The deepest feature map from the encoder is resized to $64 \times 64$, then reshaped to match the channel dimensions required by the memory-attention mechanism. Following the cross-attention operation, the refined feature map is resized back to its original resolution and passed to the decoder. The

mask for the first frame is generated using the base model, skipping the memory attention module. To ensure consistency, we reuse SAM2's positional encoding.

**Feature Alignment, Memory Encoder and Module.** For multi-class segmentation, a single-channel mask is generated by inverting the background probabiliy. The memory encoder produces a $64 \times 64$ tensor, reshaped to (B,H×W,C). At each new frame, memory representations from the 10 most recent frames are concatenated along the batch dimension. This aggregated memory is stored internally and utilized by the memory-attention mechanism to integrate contextual information from prior frames.

## 3   Experiments

### 3.1   Experimental Setup

**SAR-RARP50.** This is a publicly available dataset. We adopt the data split as described in the original SAR-RARP50 paper[10]. This dataset contains surgical videos with nine instrument classes, annotated every 60 frames.

**CholecSeg8k.** This is a dataset consists of 8,080 pixel-wise annotated frames extracted from 17 laparoscopic cholecystectomy videos in the Cholec80 dataset. Each frame includes semantic segmentation masks for 13 anatomically and surgically relevant classes, supporting detailed analysis of endoscopic scenes.[4]

**Training Settings.** We first trained baseline models (e.g., U-Net [12]/EfficientNet-B7 [15], U-Net/MaxViT-T [16], DeepLabV3+ [1]/ResNet-101 with RAdamSchedulefree [3]. The learning rate is set to $1 \times 10^{-4}$ for MaxViT encoders and $1 \times 10^{-3}$ otherwise. The models were trained for 75 epochs with a batch size of 16, applying TrivialAugment [9]. For TMAM fine-tuning and ablation studies, no augmentation was applied. We used a batch size of 1 and trained for three epochs, including unlabeled frames to facilitate learning of temporal relationships. Both training and fine-tuning utilized Generalized Dice Focal Loss, defined as:

$$\mathcal{L}_{\text{total}} = \mathcal{L}_{\text{GeneralizedDice}} + \mathcal{L}_{\text{Focal}} \tag{1}$$

where $\mathcal{L}_{\text{GeneralizedDice}}$ is Generalized Dice Loss[14], and $\mathcal{L}_{\text{Focal}}$ is Focal Loss[6].

**Evaluation Metrics.** We measure spatial segmentation accuracy using the per-class Dice Score, calculated for each individual image and then averaged. To assess temporal consistency, we modified the approach defined in [7]. Instead of using the original implementation, we employed RAFT [17] to predict optical flow between consecutive frames. The mask $t - 1$ is warped to frame $t$ using the predicted flow, and the mean Intersection over Union (mIoU) is computed between the warped mask and the predicted mask for frame $t$. This metric is averaged across all frames in each video. For each dataset, temporal consistency is evaluated at fps = 60 (SAR-RARP50) and fps = 25 (CholecSeg8k).

**Ablation Study** An ablation study was conducted to isolate the effect of TMAM. Using the same training settings—minimal augmentation with resizing only, batch size 1, and 3 epochs, we fine-tuned the base segmentation models both with and without the TMAM components.

**Implementation Details.** All experiments were conducted on a system equipped with an NVIDIA Quadro RTX 8000 GPU (48GB VRAM) running Ubuntu 22.04.3 LTS. The CPU is an AMD EPYC 7702P 64-Core Processor. The source Code is available at: https://github.com/JmeesInc/TMAM.

## 3.2   Results

**Table 1.** SAR-RARP50: Average Dice and average Temporal Consistency (TC) at 60 fps. * means additional training on 3 epochs for ablation study. Proposal method is indicated by (+TMAM).

| Model | Dice | TC@60fps |
|---|---|---|
| DeepLabV3+/ResNet-101 | 0.913 | 0.662 |
| DeepLabV3+/ResNet-101* | 0.907 | 0.657 |
| DeepLabV3+/ResNet-101(+**TMAM**) | **0.922** | **0.673** |
| U-Net/EfficientNet-B7 | 0.867 | 0.608 |
| U-Net/EfficientNet-B7* | 0.865 | 0.604 |
| U-Net/EfficientNet-B7(+**TMAM**) | **0.877** | **0.654** |
| U-Net/MaxViT-T | 0.872 | 0.583 |
| U-Net/MaxViT-T* | 0.882 | 0.590 |
| U-Net/MaxViT-T(+**TMAM**) | **0.923** | **0.669** |

**Table 2.** CholecSeg8k: Average Dice and average Temporal Consistency (TC) at 25 fps. * means additional training on 3 epochs for ablation study. Proposal method is indicated by (+TMAM).

| Model | Dice | TC@25fps |
|---|---|---|
| DeepLabV3+/ResNet-101 | 0.844 | 0.645 |
| DeepLabV3+/ResNet-101* | 0.842 | 0.645 |
| DeepLabV3+/ResNet-101(+**TMAM**) | **0.873** | **0.686** |
| U-Net/EfficientNet-B7 | 0.869 | 0.608 |
| U-Net/EfficientNet-B7* | 0.872 | 0.609 |
| U-Net/EfficientNet-B7(+**TMAM**) | **0.877** | **0.647** |
| U-Net/MaxViT-T | 0.867 | 0.574 |
| U-Net/MaxViT-T* | 0.866 | 0.572 |
| U-Net/MaxViT-T(+**TMAM**) | **0.872** | **0.577** |

Table 1 summarizes the experimental results for the SAR-RARP50 dataset across three base model configurations. In each configuration, incorporating TMAM improves the Dice Score by approximately 1 to 4 percentage points on average compared to the corresponding 2D model. TMAM also consistently enhances temporal consistency, yielding an improvement of around 15 percentage points for the first two models equipped with U-Net. Notably, temporal consistency improvements are observed.

Table 2 shows the results for the CholecSeg8k dataset. While the Dice Score improvements with TMAM are more limited depending on the model, TMAM consistently leads to performance gains across configurations.

The inference speed was not significantly different from SAM2, ran at 4 FPS on our system. The slowest configuration (ResNet+DeepLabV3+) ran at 2 FPS, while the fastest (MaxViT-UNet++) achieved 9 FPS.

### 3.3   Qualitative Evaluation

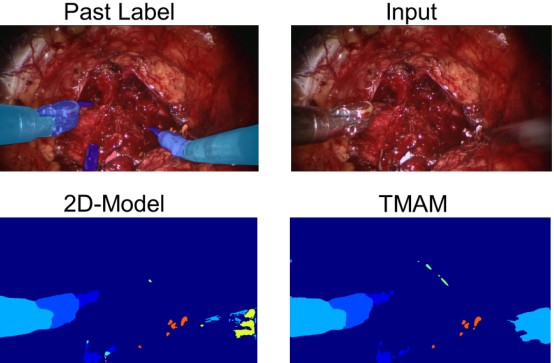

**Fig. 2.** Comparison under extreme motion. TMAM leverages temporal context to accurately segment the heavily blurred input frame (upper right). The ground truth (upper left) is from a previous, clearer frame. Sample: Video 11-1, index 180 (GT) & 210 (input).

In addition to quantitative metrics, we conducted a qualitative analysis in challenging scenarios. First, we examine cases where single-frame information is severely compromised. In Figure 2, extreme motion blur significantly degrades the input image. In Figure 3, the input image has extreme dark regions. In both cases, while 2D model fails to capture the accurate contour and instance classes, TMAM successfully reconstructs the scene and delivers a precise segmentation.

Next, we evaluate the crucial aspect of temporal consistency, especially during events like occlusion and partial visibility. Figure 4 illustrates an occlusion scenario where an instrument is temporarily hidden. While 2D models lose track of such objects, TMAM retains the instrument class once it reappears. Figure 5

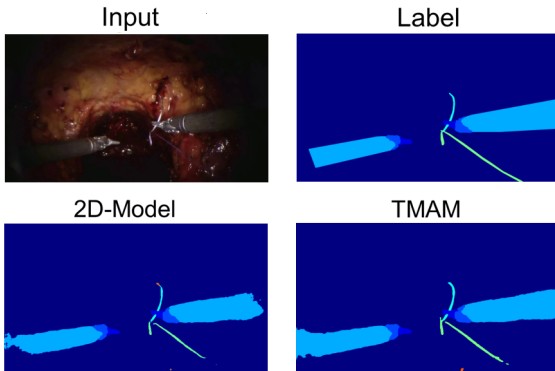

**Fig. 3.** Comparison in a low-light scenario. TMAM accurately segments the instrument in the dark, where the baseline model fails. Sample: Video 2, index 960.

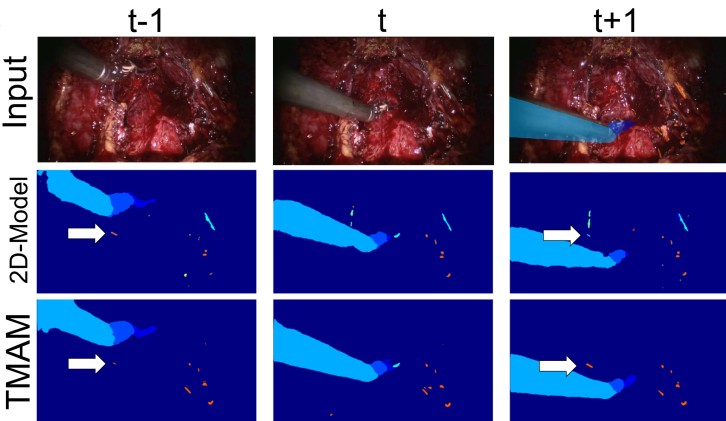

**Fig. 4.** Temporal consistency during an occlusion. The white arrow indicates the instrument hidden at frame $t$. TMAM retains the object's memory, ensuring consistent segmentation at frame $t + 1$. Sample: Video 36, indices 120, 180, 240.

further demonstrates this capability in a scenario with limited visibility, where TMAM correctly identifies an instrument whose key features are off-camera by recalling its appearance from memory.

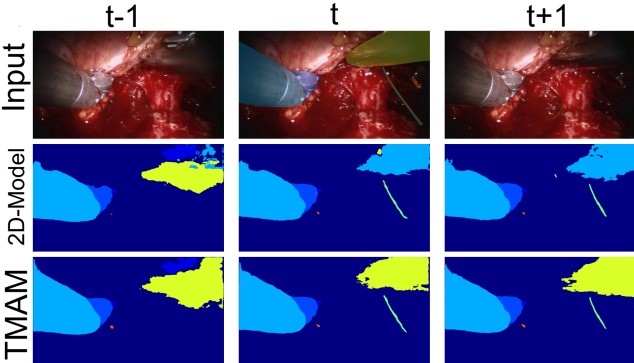

**Fig. 5.** Temporal consistency with limited visibility. TMAM correctly identifies the partially visible instrument at frame $t$ by recalling its appearance from memory. Sample: Video 7, indices 1500, 1560, 1620.

# 4    Discussion and Conclusion

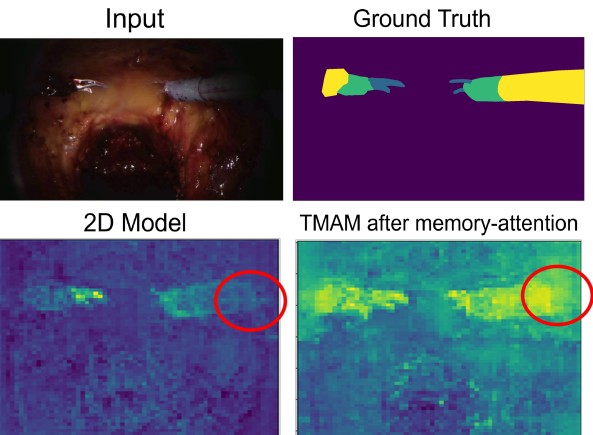

**Fig. 6.** Comparison of feature map between 2D model and TMAM.

Analysis of the feature maps in Figure 6 further demonstrates that the memory-attention mechanism compensates for critical temporal features often

overlooked by single-frame models. Comparing the input image, feature maps from a conventional 2D model, and those before and after TMAM's memory-attention suggests that TMAM enriches feature representations by integrating information from past frames.

Despite these promising results, the computational cost of TMAM remains high. Future work should explore strategies to reduce this complexity. Ensuring real-time capability is crucial for clinical applications; therefore, optimizing the inference pipeline to minimize latency while maintaining accuracy is a critical direction for further research.

In conclusion, our study demonstrates that TMAM significantly enhances both spatial segmentation accuracy and temporal consistency across different base models and datasets. These improvements are particularly pronounced in multi-class surgical tool segmentation, where zero-shot methods fall short, and in high-difficulty tasks. These advancements pave the way for more reliable and robust segmentation systems in medical video analysis, ultimately contributing to enhanced surgical precision and patient safety.

**Supplementary Material** A video used for validation with, TMAM inference, base model inference, and ground truth is available from github repository.

**Disclosure of Interests** Shunsuke Kikuchi received financial support for a research internship at Jmees Inc. Atsushi Kouno is an employee of Jmees Inc. Hiroki Matsuzaki is the co-founder and CEO of Jmees Inc.

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
