# OpenReview forum: "Memory-Enhanced Temporal Learning: Leveraging SAM2’s Memory Modules for Consistent Video Segmentation on Surgical Video"
_MICCAI.org/2025/Workshop/MSB_EMERGE — MSB EMERGE 2025 Conditionalrequiresmajorrevision_

### Official Review · Reviewer_bZbo · 2025-07-08

**Recommendation:** 3
**Confidence:** 3

**Clarity:**

The paper is generally clear but has some clarity issues that could be addressed with moderate revision

**Feedback:**

- Evaluate Efficiency: Include quantitative measurements of GPU memory usage and frame processing throughput to assess practical deployment feasibility.

- Broaden Domain Testing: Validate TMAM on natural video benchmarks and other medical imaging modalities (e.g., ultrasound, CT) to demonstrate generality.

- Clarify Initialization: Clearly specify the strategy for initializing memory on the first frame and any error-correction mechanisms to prevent drift.

**Justification:**

This work presents a concise and effective module that addresses temporal inconsistency in 2D segmentation, showing substantial gains in both spatial accuracy and temporal coherence. Its plug-and-play design and automation make it highly relevant for clinical video segmentation systems. However, analyses of computational efficiency and more diverse evaluations are needed.

**Reproducibility:**

Sufficient amount of details available for reproducing the main results, and open access is provided (or promised upon acceptance) to source code and/or data

**Strengths:**

- Generality: TMAM can be seamlessly integrated into various 2D segmentation backbones (e.g., U-Net, DeepLabV3+, MaxViT) with minimal architectural changes.

- Automation: Eliminates the need for user point prompts required by SAM2, enabling end-to-end automatic video segmentation.

- Performance Improvements: On the SAR-RARP50 dataset, TMAM yields up to a 4 percentage-point increase in mean Dice and up to a 15 percentage-point boost in TC ; similar improvements are observed on CholecSeg8k.

- Robustness Analysis: Provides both quantitative and qualitative evaluations under challenging conditions (e.g., motion blur, low illumination, occlusions), demonstrating TMAM’s resilience.

**Summary:**

This paper proposes a Temporal Memory Augmentation Module (TMAM) to enforce temporal consistency in 2D segmentation models. TMAM leverages SAM2’s memory encoder and a memory-attention mechanism to integrate predictions from previous frames into the feature representations of the current frame, thereby improving mask consistency across video frames. The module is designed as a plug-and-play component compatible with any encoder–decoder architecture and enables fully automatic video segmentation without user prompts. Experiments on two public surgical video datasets (SAR-RARP50 and CholecSeg8k) demonstrate significant gains in Dice score and Temporal Consistency (TC) when TMAM is applied.

**Weaknesses:**

- Computational Overhead: Storing and processing memory from the previous 10 frames increases both memory footprint and computation time. The paper lacks detailed analysis of runtime and GPU memory usage.

- Initialization Strategy: While full automation is claimed, the initialization procedure for the first frame (i.e., how initial memory is populated) is not clearly described. Erroneous initial predictions could propagate through subsequent frames.

- Limited Dataset Diversity: Validation is performed only on two surgical video datasets, leaving questions about TMAM’s applicability to natural videos or other medical imaging modalities.

---

### Official Review · Reviewer_6b9H · 2025-07-08

**Recommendation:** 2
**Confidence:** 4

**Clarity:**

The paper is generally clear but has some clarity issues that could be addressed with moderate revision

**Feedback:**

- The first figure and its caption are not clear.
- In the memory attention submodule there might be a confusion between the terms resized (typically refers to changing feature dimensions) and reshaped (more appropriate when converting token sequences to 2D spatial formats)

**Justification:**

While the paper shows promising empirical results and offers a flexible, easily integrable module, it lacks technical novelty, relies heavily on existing components, and presents unclear or overstated claims regarding its contributions. Therefore, in its current state, the paper is not ready for publication.

**Reproducibility:**

Sufficient amount of details available for reproducing the main results, but open access is not provided to source code and/or data

**Strengths:**

- **Empirical results**: The proposed module enhances the performance of various methods across two datasets.
- **Flexibility**: The plug-and-play design allows for easy integration into different architectures.

**Summary:**

The paper presents a module designed to incorporate information from previous frames to enhance temporal consistency in endoscopic surgery video analysis. The approach uses SAM2's memory encoder and memory attention mechanisms to guide predictions in subsequent frames. Experimental results demonstrate consistent improvements across multiple architectures and two different datasets.

**Weaknesses:**

- **Limited technical novelty**: The proposed TMAM module directly reuses SAM2's memory encoder and memory attention mechanisms.
- **Unclear initialization for the first frame**: Since the method relies on predictions from previous frames to guide segmentation, it is unclear how the system initializes at frame 0 when no prior predictions are available.
- **Potentially misleading contribution claim**: The paper suggests that SAM presents a barrier for system automation. However, SAM supports various types of prompts (e.g., points, boxes, masks, text), many of which can be automatically generated. Moreover, SAM2 does not require prompts for every frame since it uses the previous predictions. Therefore, the claimed limitation of prompt usage is overstated and may not represent an actual barrier in practice.

---

### Official Review · Reviewer_sqRt · 2025-07-09

**Recommendation:** 3
**Confidence:** 4

**Clarity:**

The paper is generally clear but has some clarity issues that could be addressed with moderate revision

**Feedback:**

The authors need to provide more details of their experiments and more ablation is needed

1. The paper needs to provide a clear idea of how the SAM 2 was adapted for TMAM. The idea makes sense at a high level, but the details of implementation needs to be provided. What is the role of the feature alignment modules ? Since the inout and output are pixel aligned - what does a separate alignment module achieve ? How does it effect the accuracy (without TMAM).
2. The authors need to analyze the edge cases like blur and rapid motion which are more likely to effect a method that contains temporal feedback module like TMAM than image-by-image techniques. It can be hinderance to potential applications if the method does not perform well on these edge cases
3. Minor area of improvement - use diverse datasets (other than surgical videos). This can easily make this paper stronger by showing applications across domains

**Justification:**

The paper presents a promising adaptation of the SAM 2 architecture for video segmentation but does not provide sufficient analytical details beyond just DICE and temporal consistency scores. Video supplementary data, details of implementation, etc are missing. The paper wastes a lot of space on since centered figures, which can be easily used for more analysis.

**Reproducibility:**

Some amount of details available for reproducing the main results, and open access details are unclear

**Strengths:**

1. The paper shows consistent improvement across the 2 datasets and 3 models used as base models. This shows that the introduction of temporal attention improves the DICE score and temporal consistency metrics across the board
2. The authors performs ablation studies to eliminate the some effect of new training introduced by the TMAM fine-tuning.

**Summary:**

In this paper, the authors propose a way to adapt an encoder decoder based image segmentation network to be able to take into account temporal context. The authors propose TMAM (Temporal Memory Augmentation Module) which is an adaptation of the memory bank and memory attention backbone from SAM 2 segmentation model. Here the authors selectively retrain parts of the segmentation pipeline on surgical scene video datasets like SAR-RARP50 and CholecSeg8k. The authors show that using TMAM to adapt different base segmentation networks show improvement over the ones without TMAM.

**Weaknesses:**

1. The paper does not talk about the hard cases for video segmentation. For example - what happens when there are rapid motions in the video ? These are considered to the hard cases for temporal attention networks, but does not effect frame-by-frame segmentation. Since only overall metrics are used for evaluation, these limitations can be hidden by performance in continuous videos, which is majority of the dataset.
2. While the authors do show a few image samples, since this is video segmentation paper, it would be nice to see the video segmentation results in the supplementary section.
3. The paper talks about the feature alignment steps before and after the memory attention in figure 1, but provide no details about what this step does and is achieved in practice. This not only makes the method less reproducible, but also raises questions around if TMAM by itself improves performance - or the improvements come from better alignment. An ablation study here would be needed
4. The paper’s title as well as the implementation suggests that the method is capable of generalizing to any video segmentation task and not just surgical videos. However, the authors only show the performance on two datasets which are both surgical video datasets. It would be nice to see if this method generalized to other domains

---

### Decision · Program_Chairs · 2025-07-18

**Decision:**

Conditional Accept (requires major revision)

**Comment:**

The paper is conditionally accepted to the EMERGE workshop. While the reviewers recognize the value of the contribution, the claims and title should be revised to accurately reflect the narrow scope of the application. We ask the authors to clearly state the limitations and adjust the presentation of novelty accordingly in the camera-ready version.

Acceptance is contingent on satisfactorily addressing these concerns.